# Scopoletin Reactivates Latent HIV-1 by Inducing NF-κB Expression without Global T Cell Activation

**DOI:** 10.3390/ijms241612649

**Published:** 2023-08-10

**Authors:** Yuqi Zhu, Zhengtao Jiang, Lin Liu, Xinyi Yang, Min Li, Yipeng Cheng, Jianqing Xu, Chunhua Yin, Huanzhang Zhu

**Affiliations:** 1State Key Laboratory of Genetic Engineering and Engineering Research Center of Gene Technology, Ministry of Education, Institute of Genetics, School of Life Sciences, Fudan University, Shanghai 200438, China; 13761909553@163.com (Y.Z.); 16110700035@fudan.edu.cn (Z.J.); liulin@aurisco.com (L.L.); xinyy@fudan.edu.cn (X.Y.); 21210700031@m.fudan.edu.cn (M.L.); 22210700105@m.fudan.edu.cn (Y.C.); chyin@fudan.edu.cn (C.Y.); 2Scientific Research Center, Shanghai Public Health Clinical Center, Fudan University, Shanghai 201508, China; xujianqing@shphc.org.cn

**Keywords:** Scopoletin, HIV-1 latency, reactivation, NF-κB, T-cell activation

## Abstract

Reversing HIV-1 latency promotes the killing of infected cells and is essential for cure strategies. However, current latency-reversing agents (LRAs) are not entirely effective and safe in activating latent viruses in patients. In this study, we investigated whether Scopoletin (6-Methoxy-7-hydroxycoumarin), an important coumarin phytoalexin found in plants with multiple pharmacological activities, can reactivate HIV-1 latency and elucidated its underlying mechanism. Using the Jurkat T cell model of HIV-1 latency, we found that Scopoletin can reactivate latent HIV-1 replication with a similar potency to Prostratin and did so in a dose- and time-dependent manner. Moreover, we provide evidence indicating that Scopoletin-induced HIV-1 reactivation involves the nuclear factor kappa B (NF-κB) signaling pathway. Importantly, Scopoletin did not have a stimulatory effect on T lymphocyte receptors or HIV-1 receptors. In conclusion, our study suggests that Scopoletin has the potential to reactivate latent HIV-1 without causing global T-cell activation, making it a promising treatment option for anti-HIV-1 latency strategies.

## 1. Introduction

Antiretroviral therapy (ART) is effective in inhibiting the replication of HIV-1, leading to a decline in viral loads to clinically undetectable levels [1]. However, the complete eradication of the virus remains challenging due to the existence of latent reservoirs comprising HIV-1 latently infected cells [2,3,4]. Multiple mechanisms contribute to the formation of latent reservoirs, including the suppression of transcription factors such as nuclear factor kappa B (NF-κB), nuclear factor of activated T cells (NFAT), activating protein-1 (AP-1), and specificity protein 1 (SP1) [5,6,7,8,9]. Another factor contributing to HIV latency is the positive transcriptional elongation factor b (P-TEFb), which impedes efficient transcription elongation from the HIV-1 promoter [10,11]. Epigenetic modifications acting on viral promoter long terminal repeats (LTR) can also contribute to the formation of viral reservoirs [12,13]. Additionally, posttranscriptional restrictions and miRNAs may help maintain HIV-1 latency [14,15,16,17].

In light of the molecular understanding of HIV-1 latency, a therapeutic strategy called “shock and kill” has been developed with the aim of curing HIV [18]. This strategy involves reactivating latent HIV-1 reservoirs using latency-reversing agents (LRAs), followed by the immune clearance of the virus-infected cells [11,19,20].

Epigenetic LRAs target the chromatin restrictions that maintain HIV latency and include histone deacetylase inhibitors [21,22,23,24], histone methyltransferase inhibitors [25], and bromo- and extra-terminal domain inhibitors [26,27,28]. Signal agonist LRAs stimulate immune signaling pathways in T cells, leading to transcriptional modifications and NF-κB activation. These LRAs include PKC agonists [29], cytokines [30,31,32], Toll-like receptor agonists [33,34,35], mimetics of the second mitochondrial-derived activator of caspases [36], and immune checkpoint inhibitors [37,38]. Clinical trials using LRAs such as SAHA [24,39,40,41,42], romidepsin [43], panobinostat [23], bryostatin-1 [44], disulfiram [45], and others have shown promising results, although their impact on the sizes of latent reservoirs has been limited or not significant [30]. Developing more potent and safer LRAs is urgently needed.

Prostratin is extracted from the Samoan plant *Homalanthus nutans* and used treat yellow fever [46]. Many recent studies have shown that prostratin can also reactivate the latent HIV proviruses in infected thymocytes [47,48,49] by activating NF-κB [49].

Scopoletin is an important coumarin phytoalexin found in plants. It has been associated with several pharmacological activities, including anti-tumor, hypolipidemic, and spasmolytic effects, and agricultural biological activity [50]. We therefore investigated whether Scopoletin (6-Methoxy-7-hydroxycoumarin), which is widely found in the root of plants of genera such as Rutaceae, Compositae, and Umbelliferae [51], can reactivate latent HIV-1 in latently infected A10.6 and C11 cell lines and explored its reactivation mechanism, as well as its effect on global T-cell activation.

## 2. Results

### 2.1. Scopoletin Activates Latent HIV-1 Replication

The structure of Scopoletin is shown in Figure 1a. A10.6 and C11 cell lines, which are HIV-1 latently infected Jurkat T cell lines, were used with a green fluorescent protein (GFP) gene under the control of HIV-1 LTR [52] to easily detect the reactivation of latent HIV-1 by flow cytometry. A10.6 and C11 cells were treated with different concentrations of Scopoletin or Prostratin for 72 h. The percentage of GFP+ cells was then measured to determine the transcriptional activity of the HIV-1 promoter. It was found that the percentage of GFP+ A10.6 cells increased to 30.29% (Figure 1b), and that of GFP+ C11 cells increased to 40.09% (Figure 1c).

As shown in Figure 2a,b, the percentage of GFP+ cells was positively correlated with the concentration of Scopoletin. When the concentration of Scopoletin increased from 0.2 to 1.6 mM, the percentage of GFP+ A10.6 cells rose from 10.72% to 28.87% (Figure 2). When the concentration of Scopoletin increased from 0.25 to 2 mM, the percentage of GFP+ C11 cells rose from 8.02% to 42.65% (Figure 2b). Similar results were observed following treatment with Prostratin. These results confirmed that Scopoletin induced HIV-1 LTR reactivation in a dose-dependent manner.

In order to analyze the kinetics of the Scopoletin-induced HIV-1 LTR expression, a kinetics experiment was performed. As shown in Figure 2c, after A10.6 cells were treated with Scopoletin, the percentage of GFP+ cells increased in a time-dependent manner. Seventy-two hours after treatment, the percentage of GFP+ cells reached its highest level. In C11 cells, the kinetics of the Scopoletin-induced HIV-1 LTR expression increased rapidly for the first 3 days and more steadily by day 4 (Figure 2d). These results indicated that Scopoletin can activate latent HIV-1 transcription in a time-dependent manner. Meanwhile, an ELISA assay was performed to measure the P24 antigen expressed by the latently infected cell line. The results showed that A10.6 expressed the P24 antigen after activation by both Scopoletin and Prostratin, and similar results were obtained (Figure 2e). Among these results, Scopoletin could reactivate latent HIV-1 and was therefore selected as a candidate for further investigation.

### 2.2. Testing the Synergistic Activation of HIV-1 Production by Scopoletin and Other Activators

When the combination of two drugs produces a greater activation level than the sum of the activations produced by individual activators, the effect is considered synergistic [53]. To investigate whether Scopoletin synergistically reactivates the HIV-1 promoter with VPA, 5-Aza, or TNF-α, A10.6 cells were mock-treated or treated with Scopoletin (2.0 mM), VPA (2.0 mM), TNF-α (10 ng/mL), 5-Aza (100 nM), Scopoletin (2.0 mM)/TNF-α (10 ng/mL), Scopoletin (2.0 mM)/5-Aza (100 nM), or Scopoletin (2.0 mM)/VPA (2.0 mM) for 48 h. As shown in Figure 3a, stimulation with Scopoletin, VPA, or 5-Aza alone induced GFP expression in a small percentage of cells (23.83, 6.76, and 1.82%, respectively). In contrast, TNF-α alone induced 53.9% of cells to express GFP. When cells were co-treated with a combination of Scopoletin and the other drugs, a higher percentage of GFP+ cells was observed (29.29%, 17.93%, and 70.54%, respectively). These results indicated that combination treatment with Scopoletin and VPA, 5-Aza, or TNF-α has almost no synergistic activity in the reactivation of HIV-1 production.

### 2.3. Cytotoxicity Analysis of Scopoletin In Vitro

To test the potential cytotoxicity of Scopoletin, human primary PBMCs, Jurkat T and HEK 293 cells were treated with different concentrations of Scopoletin. An assay based on the conversion of a tetrazolium salt to a formazan product (CCK-8 assay) was conducted to measure the cell viability. Scopoletin did not exhibit cytotoxicity in human primary PBMCs at any of the different concentrations (Figure 3b). A significant correlation was found between the concentration of Scopoletin and the viability of HEK 293 and Jurkat T cells (Figure 3c,d). The 50% cytotoxic concentration (CC50) of Scopoletin was calculated in these cells to further evaluate cytotoxicity. The CC50 in the HEK 293 and Jurkat T cells for Scopoletin was 15.18 and 17.366 mM, much higher than the working concentration, indicating that it is very safe at its active concentration.

Currently, another major side effect of certain therapeutic agents is their propensity to non-specifically activate global T cells. The induction of T cell activation markers was therefore tested. The expression of CD25 and CD69 was detected following treatment with Scopoletin in PBMCs from healthy donors. Prostratin stimulated a high expression of these activation markers, as previously reported [54] (Figure 4a). Flow cytometric analysis showed that Scopoletin did not significantly induce the expression of CD25 or CD69, as compared to Prostratin (Figure 4b,c).

The cell surface expression of HIV-1 receptors/co-receptors is important for viral attachment and entry into immune cells [55]. SAHA, known to efficiently reactivate latent HIV, was reported to increase the susceptibility of naive CD4+ T cells to HIV-1 acquisition [56]. On the contrary, prostratin and analogs may have protective effects against HIV-1 infection [54,57]. We sought to examine the effect of Scopoletin on the expression of cell surface HIV-1 receptors/co-receptors. Human PBMCs from healthy donors were treated with Scopoletin for 72 h. The treated PBMCs were then evaluated for the expression of CD4, CCR5, and CXCR4 using flow cytometry. The data showed that Scopoletin did not increase the expression of HIV-1 receptors/co-receptors, suggesting that Scopoletin may not increase the susceptibility of PBMCs to HIV infection (Figure 5).

In summary, treatment with Scopoletin had no significant impact on cell viability, T cell activation biomarkers, or HIV-1 receptors/co-receptors. These results indicated that Scopoletin has a low cytotoxicity at its active concentration. This should be further evaluated in animal models, which is critical in the clinical research of potential LRAs.

### 2.4. Scopoletin-Mediated Activation of HIV-1 Involves the Induction of NF-κB Expression

Advancements have been made in our understanding of the mechanisms of HIV-1 latency, mostly due to the reduced transcriptional activity of the viral promoter LTR. Several binding sites for transcription factors were found in the HIV-1 LTR, including NF-κB, AP-1, and SP1 [58]. To determine which factor Scopoletin’s reactivation uniquely depends on, HEK-293 cells were transfected with luciferase reporter plasmids containing the wild-type HIV-1 LTR, LTR lacking the two κB enhancers, LTR lacking the AP-1 enhancers, or LTR lacking the SP1 enhancers, followed by treatment with 2 mM Scopoletin or 10 ng/mL TNF-α or 1 uM Prostratin. TNF-α was chosen here as a positive control. Scopoletin induced a ~four-fold stimulation of the HIV-LTR-Luc reporter relative to the mock controls. However, it failed to activate the HIV-LTR (ΔκB)-luciferase reporter. TNF-α and Prostratin also failed to activate the HIV-LTR (ΔκB)-luciferase reporter (Figure 6a). Furthermore, Scopoletin (2.0 mM) and TNF-α (10 ng/mL) were used to treat wild-type HIV-1 LTR-luc-neo cells, which express luciferase by controlling intact LTR, and mutant HIV-1 LTR ΔκB-luc-neo cells; NF-κB was knocked out. At 48 h post-treatment, the cells were lysed, and luciferase activity was measured. The results showed that wild-type HIV-1 LTR-luc-neo cells can be activated by Scopoletin. However, the ratio of GFP+ cells does not increase in HIV-1 LTR ΔκB-luc-neo cells (Figure 6b). These results suggested that NF-κB may be involved in the Scopoletin-mediated activation of HIV-1 LTR.

To further confirm the role of NF-κB in Scopoletin-mediated stimulation, C11 cells were treated with multiple concentrations of Aspirin (acetyl salicylic acid) for 3 h and then Scopoletin (2 mM) or Prostratin (10 ng/mL) for 24 h. Aspirin can inhibit the Prostratin-induced activation of NF-κB [59]. In the present study, Aspirin not only inhibited the Prostratin-induced GFP expression but also significantly reduced the activation by Scopoletin (Figure 6c). These data further encourage our conclusion that NF-κB was involved in the Scopoletin-mediated activation of latent HIV.

In addition, we investigated whether the Scopoletin stimulation caused NF-κB nuclear translocation and DNA binding by EMSA. We hypothesized that this could shed some light on the role of NF-κB in Scopoletin-mediated stimulation. Following the treatment of C11 cells with Scopoletin or TNF-α, nuclear extracts were collected. They were then incubated with NF-κB enhancer oligonucleotides. C11 cells treated with Scopoletin exhibited a dose-dependent induction of NF-κB DNA binding, as compared to the mock control. Pretreatment with aspirin suppressed the induction of NF-κB DNA binding in both the Scopoletin and TNF-α groups (Figure 6d). In combination, the present results showed that the reactivation of latent HIV-1 by Scopoletin mainly involves the NF-κB pathway.

## 3. Discussion

Natural products derived from microbes and medicinal plants have played an important role in the discovery and development of drugs. In the present study, we investigated the effect of Scopoletin, which has been associated with several pharmacological activities, including anti-tumor, hypolipidemic, and spasmolytic effects, as well as agricultural biological activity. It was found that Scopoletin can effectively reactivate latent HIV-1 in different latency models. Of note, the MTT assay results showed that Scopoletin was not cytotoxic in PBMCs. However, it is important to extend these observations to primary latency models or latent cells from infected individuals undergoing ART to confirm whether Scopoletin is a potential drug candidate in anti-HIV-1-latency therapy. Furthermore, we investigated the synergistic effect of Scopoletin with 3 other LRAs to achieve a higher efficiency and lower cytotoxicity. Regrettably, the data showed almost no synergistic activity in different concentrations. This was possibly due to the fact that the mechanisms of selected activators are similar to Scopoletin’s in reversing HIV-1 latency. Further experiments are required to explore the synergistic effect of Scopoletin with other anti-latency agents.

Global T cell activation can have deleterious effects. Therefore, a safe LRA would reactivate latent HIV-1 with minimized cytotoxicity and no global T cell activation. We assessed the effect of Scopoletin on T lymphocyte activation biomarkers. Flow cytometry showed that Scopoletin has no stimulatory effect on T lymphocyte activation biomarkers. Furthermore, Scopoletin has almost no effect on HIV-1 receptors/co-receptors.

To investigate the major mechanism of Scopoletin in the activation of HIV-1, HEK-293 cells were transfected with HIV-LTR (ΔκB/ΔAP-1/ΔSP-1)-luciferase reporter plasmids. The cells were then treated with Scopoletin and measured by a Luciferase assay. The data showed that the expression of the HIV-LTR (ΔκB) group decreased dramatically. Simultaneously, we studied Scopoletin’s activation effect in in vitro models, which were established by luciferase stable expression cell lines driven by wild-type or mutant LTR with the κB element deleted. The luciferase assay results indicated that the activation effect in the ΔκB cell line was clearly lower than that in the wild-type LTR. These results suggested that Scopoletin can influence LTR through the NF-κB signaling pathway. We then investigated its stimulation effect when combined with Aspirin, a reported NF-κB inhibitor [59]. C11 cells were pretreated with Aspirin, followed by treatment with Scopoletin or Prostratin. It was found that the reactivation efficiency of both Scopoletin and Prostratin was significantly decreased, directly confirming that Scopoletin can intervene in HIV-1 latency through the NF-κB signaling pathway. Finally, EMSA was performed to detect which nucleus proteins were specifically bound to the NF-κB probe following treatment with Scopoletin alone or pretreatment with Aspirin. The results revealed a specific band of p50/p65 heterodimer binding with the NF-κB probe following treatment with Scopoletin, as compared with mock treatment. Like the positive control TNF-α, the binding band of Scopoletin’s group presented a concentration gradient, while in the group pretreated with Aspirin, a distinct reduction in or even disappearance of the specific binding band between the p50/p65 heterodimer and the NF-κB probe was observed, indicating that Scopoletin can effectively promote RelA/p65 translocation to the nucleus. The result that Scopoletin can reactivate latent HIV-1 by inducing NF-κB expression is consistent with some previous findings [60,61], but there are also reports that this compound, as anti-inflammatory activity, cannot activate the NF-κB pathway [62,63,64]. The conflicting results may be related to different cell lines and drug doses.

An ideal LRA will be potent, nontoxic, non-cell-activating, and unable to impair the cytotoxic effector function of CTLs and NK cells [65]. With a description of reactivators described by Gramatica et al., they identified a new class of LRAs including two glycogen synthase kinase-3 inhibitors (GSK-3is), SB-216763, and tideglusib, which activate AKT/mTOR signaling. These GSK-3is reactivated latent HIV-1 present in blood samples from aviremic individuals on antiretroviral therapy (ART) in the absence of T cell activation, the release of inflammatory cytokines, cell toxicity, or an impaired effector function of cytotoxic T lymphocytes or NK cells [65]. It would be of interest to directly compare the effects of Scopoletin and the GSK-3 inhibitors in the Jurkat-derived cell lines and in patients’ samples.

## 4. Materials and Methods

### 4.1. Cell Culture and Reagents

The A10.6 cell line (ARP-9849 Human T Lymphocyte) is a clonal HIV latently infected Jurkat T cell line [66]. The A10.6 cell line was obtained from the NIH AIDS Research and Reference Reagent Program. A10.6 cells contain a full-length integrated HIV genome with the *GFP* in place of the *Nef* and a frameshift mutation in the *env* gene [66]. The cell line C11, also derived from Jurkat cells, expresses GFP as a marker of Tat-driven HIV LTR activity and carries mutations in the *Env* and *Vpr* genes. They were constructed in our lab and used elsewhere [16,52,66,67,68,69]. C11 and A10.6 cells were grown in suspension in RPMI 1640 medium plus 10% FBS (Gibco; Thermo Fisher Scientific, Waltham, MA, USA), streptomycin (100 µg/mL) and penicillin (100 U/mL) (Invitrogen; Thermo Fisher Scientific, MA, USA) at 37 °C in a 5% CO_2_ humidified atmosphere. Scopoletin was purchased from Sigma-Aldrich (Merck KGaA, Darmstadt, Germany). Prostratin was purchased from LC laboratories (Woburn, MA, USA). 5-azacytidine (5-Aza) and Aspirin were purchased from Sigma-Aldrich (Merck KGaA). VPA was purchased from InvivoGen (San Diego, CA, USA). Recombinant human tumor necrosis factor-α (TNF-α) was purchased from Chemicon International (Thermo Fisher Scientific, Inc.). Scopoletin (1 M), TNF-α (10 µg/mL), and prostrain (1 mM) were dissolved in anhydrous dimethyl sulfoxide and stored at −20 °C.

### 4.2. Isolation of Human Peripheral Blood Mononuclear Cells (PBMCs)

White blood cells from healthy donors were purchased from the Blood Center of Shanghai (Shanghai, China). This study was approved by the Ethics Committee of Fudan University, and the methods were consistent with the relevant guidelines and regulations of that committee. PBMCs were collected by Ficoll-Hypaque gradient separation (GE Ficoll-Paque PLUS; GE Healthcare, Marlborough, MA, USA). The PBMCs were cultured in RPMI 1640 medium containing 10% FBS (Gibco; Thermo Fisher Scientific, Inc.), 100 U/mL of penicillin, and 100 µg/mL of streptomycin (Invitrogen; Thermo Fisher Scientific) at 37 °C in an incubator containing 5% CO_2_.

### 4.3. Flow Cytometry

C11 and A10.6 cells were incubated with various concentrations of drugs for different times. The cells were then washed and harvested in cold PBS, and the GFP expression was measured using a BD FACScan Flow Cytometer (BD Biosciences, Franklin Lakes, NJ, USA). The cells were analyzed using Cell Quest software Version 6.1 (BD Biosciences). Live cells were gated, and two parameter analyses were used to differentiate GFP-associated fluorescence from background fluorescence. A total of 10,000 gated events were collected per sample. The data represent the percentage of GFP-expressing cells in all gated events.

The PBMCs were treated with Scopoletin or Prostratin. Next, 0.5 × 10^6^ cells were collected and stained with FITC-conjugated anti-CD25 (BD Biosciences, 560990) and PE-conjugated anti-CD69 antibodies (BD Biosciences, 560968) to analyze T cell activation. PE-conjugated Anti-CD4 (BD Biosciences, 561843), FITC-conjugated anti-CCR5 (BD Biosciences, 561747), and FITC-conjugated anti-CXCR4 antibodies (MBL D123-4) were also used to stain the treated PBMCs. The treated PBMCs were washed with PBS and incubated for 30 min on ice with 2 µL of antibodies in 100 µL of PBS. Subsequently, the cells were washed three times and harvested in PBS. A total of 10,000 gated events were collected per sample.

### 4.4. P24 Antigen ELISA Assay

The HIV Gag P24 DuoSet ELISA kit (R&D, DY3760-05, Minneapolis, MN, USA) was used to detect the P24 antigen expressed with reactivated cells. The plate coated by a capture antibody was incubated overnight at room temperature. Briefly, the coated plate was blocked with diluted 10× the Reagent Diluent concentrate for 1 h at room temperature. After washing, the cell lysate samples were added and incubated for 2 h at room temperature. The plates were then washed, and the Detection Antibody was added. After rinsing, as mentioned above, the Streptavidin-HRP (Horseradish Peroxidase) B was added as a secondary antibody and incubated for 20 min in the dark. With the Substrate Solution (Color Reagent A: Colo Reagent B = 1:1), it was tested in the microplate reader (Biotek synergy 2, Winooski, VT, USA) after 20 min.

### 4.5. Cytotoxicity Assay

A Cell Counting Kit-8 (CCK-8) (Dojindo Molecular Technologies, Rockville, MD, USA) was used to assess the cytotoxicity of Scopoletin. The amount of the formazan dye generated by the activity of dehydrogenases in cells is directly related to the number of living cells. According to the protocol, the cells were seeded in a 96-well plate at a density of 40,000 cells per well with Scopoletin for 48 or 96 h. Next, 10 µL of CCK-8 solution was added in each well. Following incubation for 4 h at 37 °C, absorbance was measured at a wavelength of 450 nm using a microplate reader (Biotek synergy 2).

### 4.6. Transient Transfection and Luciferase Assays

The HIV-LTR-luciferase reporter was obtained from Dr. Warner C. Greene [70]. HIV-LTR (ΔκB)-luciferase reporter, HIV-LTR (ΔAP-1)-luciferase reporter, and HIV-LTR (ΔSP-1)-luciferase reporter were obtained from Dr. Andrew D. Badley [71]. The day prior to transduction, HEK-293 cells were seeded at a density of 1 × 10^5^ cells per well in 24-well plates. Twenty-four hours later, it was transfected with plasmids (0.8 μg) using Lipofectamine 2000, following the manufacturer’s instructions (Invitrogen; Thermo Fisher Scientific). After 24 h, the cells were treated with Scopoletin (2 mM), TNF-α (10 ng/mL), and Prostratin (1 μM). The cells were lysed after 48 h, and the luciferase activity was measured using a Dual-Luciferase Reporter Assay Kit (Promega Corporation, WI, USA).

### 4.7. Cell Nuclear Protein Extraction and Electrophoretic Mobility Shift Assay

C11 cells were pretreated with Aspirin (10 mM) for 1 h and subsequently treated with Scopoletin (0.5, 1, and 2 mM) or TNF-α (10 ng/mL) for 3 h. Nuclear extracts were isolated and subjected to an electrophoretic mobility shift assay with biotin-labeled NF-κB enhancer DNA probes, as previously described [69,72].

### 4.8. Statistical Analysis

The data are presented as the mean ± standard error and were calculated from at least three independent experiments performed in triplicate. Statistical significance was determined using the two-way Student *t* test. *p* < 0.05 was considered to indicate a statistically significant difference. * *p* represented a *p*-value less than 0.05, ** *p* represented a *p*-value less than 0.01, and **** *p* represented a *p*-value less than 0.0001.

## 5. Conclusions

In conclusion, the aim of the present study was to investigate the ability of Scopoletin to induce the expression of latent HIV in HIV-1 latently infected cell models. The results indicated that Scopoletin can markedly intervene in HIV-1 latency without cytotoxicity. We further verified that its reactivation mechanism involves the NF-κB signaling pathway, as well as RelA/p65 nuclear translocation. These findings suggested that Scopoletin could potentially be used as an anti-latency medicine in HIV-1 therapy.

## Figures and Tables

**Figure 1 ijms-24-12649-f001:**
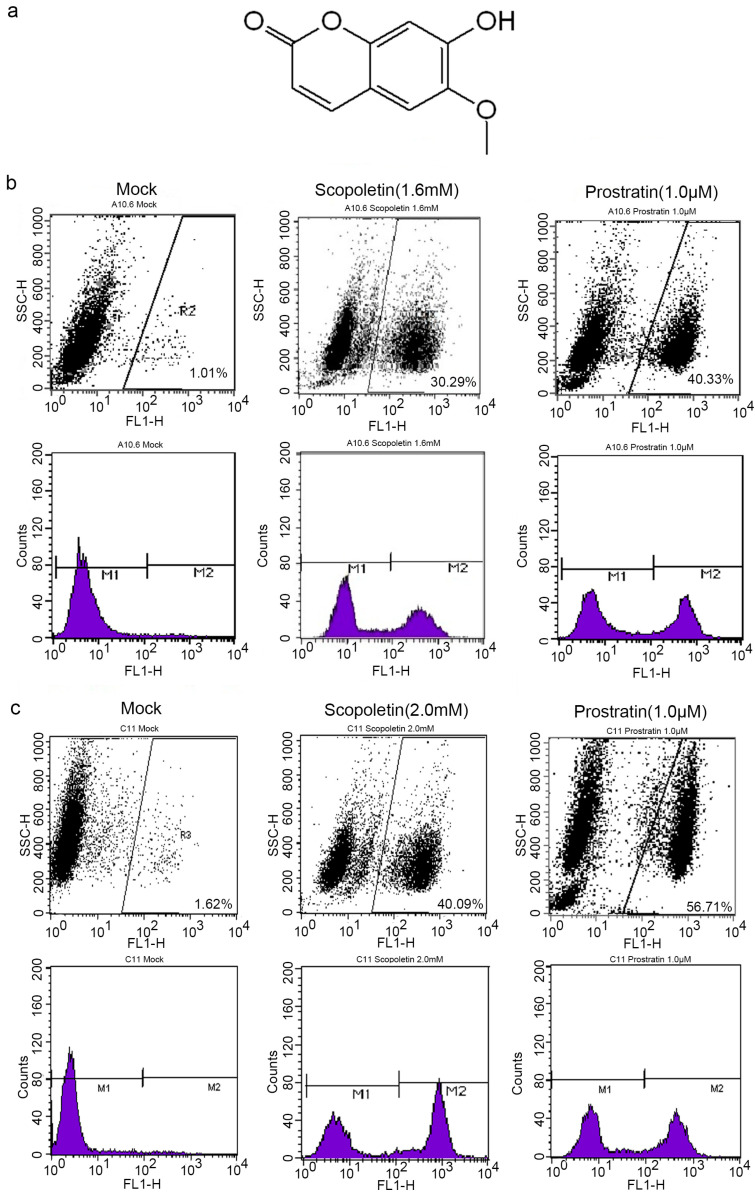
Reactivation of latent HIV-1 in latently infected C11 and A10.6 cells by Scopoletin and Prostratin. (**a**) Structure of Scopoletin. (**b**) A10.6 cells treated with Scopoletin (1.6 mM) or Prostratin (1.0 µM) for 72 h. The percentage of GFP+ cells was measured by flow cytometry to evaluate the GFP expression. The results are presented as fluorescence histograms. (**c**) C11 cells treated with Scopoletin (2.0 mM) or Prostratin (1.0 µM) for 72 h.

**Figure 2 ijms-24-12649-f002:**
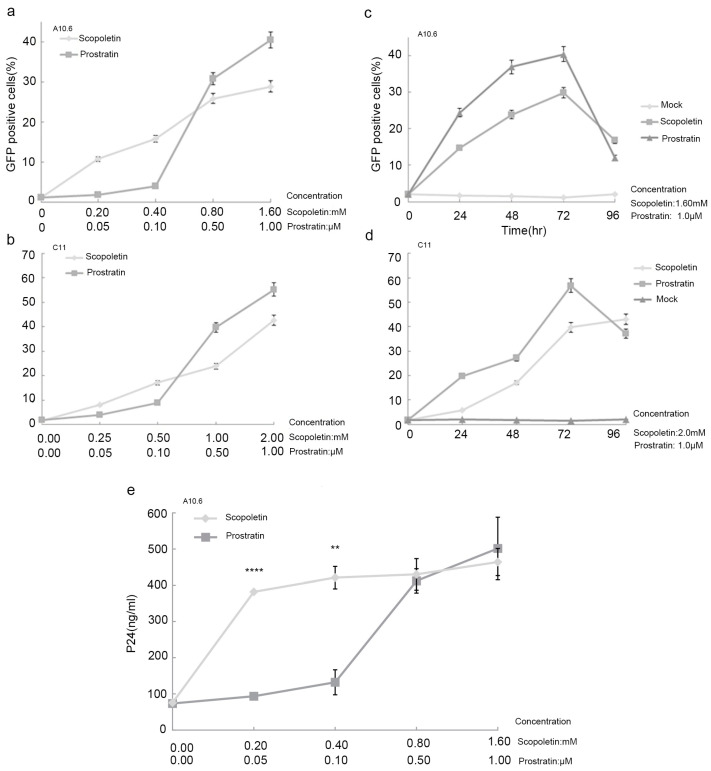
The reactivation effect of Scopoletin on A10.6 or C11 cells for different concentrations and times. (**a**) Dose-dependent effects of Scopoletin on HIV-1 production in A10.6 cells. Data are presented as the mean ± standard deviation of three independent experiments. (**b**) Dose-dependent effects of Scopoletin in C11 cells. (**c**) A10.6 cells were treated with Scopoletin (1.6 mM) or Prostratin (1.0 µM). At the indicated times, the data are expressed as the percentage of GFP+ cells. (**d**) Time-dependent effects of Scopoletin on C11 cells. (**e**) A10.6 cells expressed P24 antigen after being activated by Scopoletin and Prostratin for 72 h. Data are presented as the mean ± standard deviation of three independent experiments. ** *p* < 0.01, **** *p* < 0.0001.

**Figure 3 ijms-24-12649-f003:**
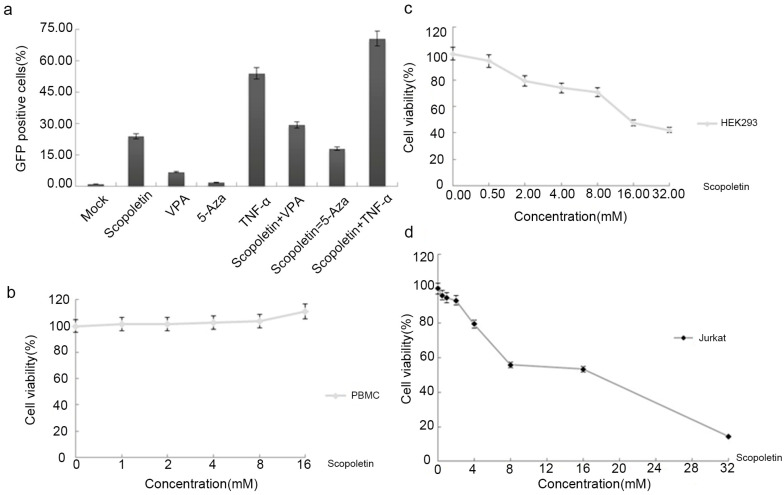
Testing of the synergistic activation of HIV-1 production by Scopoletin and other activators and analysis of Scopoletin cytotoxicity in vitro. (**a**) Synergistic activation of HIV-1 promoter by Scopoletin and other activators in latently infected cells. A10.6 cells received either mock treatment or treatment with Scopoletin (2.0 mM), VPA (2.0 mM), TNF-α (10 ng/mL), 5-Aza (100 nM), Scopoletin (2.0 mM)/TNF-α (10 ng/mL), Scopoletin (2.0 mM)/5-Aza (100 nM), or Scopoletin (2.0 mM)/VPA (2.0 mM) for 48 h, respectively. A10.6 cells were analyzed by flow cytometry. HIV-GFP reactivation is reported as the percentage of GFP+ cells. Data are presented as the mean ± standard deviation of three independent experiments. (**b**) Human PBMCs were treated with Scopoletin at the indicated concentrations for 48 h and measured by the CCK-8 method. (**c**) HEK 293 cells were treated with Scopoletin at the indicated concentrations for 72 h and then measured. (**d**) Jurkat T cells were treated with Scopoletin at the indicated concentrations for 72 h and then measured.

**Figure 4 ijms-24-12649-f004:**
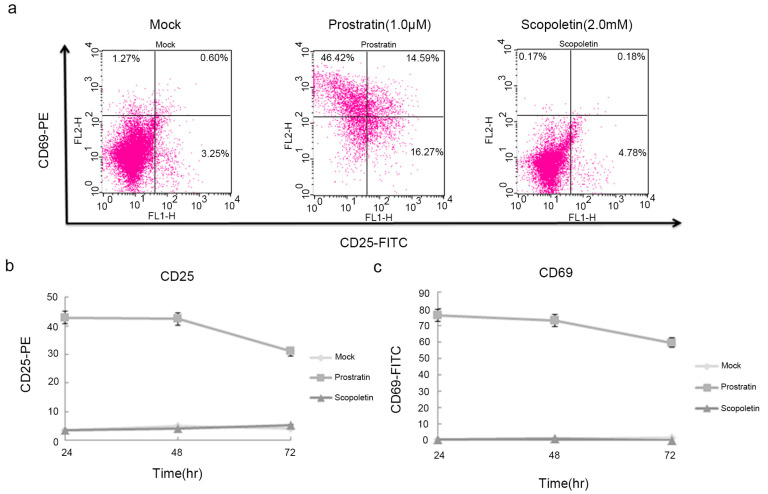
Effects of Scopoletin on the expression of CD25 and CD69. (**a**) Human PBMCs were treated with Scopoletin (2 mM) or Prostratin (1 µM) for 72 h. The expression of CD25 and CD69 was detected by flow cytometry using antibodies against CD25 and CD69, respectively. Fold change in the expression of (**b**) CD25 and (**c**) CD69, as compared to mock treatment for different timepoints. The results are representative of three independent experiments.

**Figure 5 ijms-24-12649-f005:**
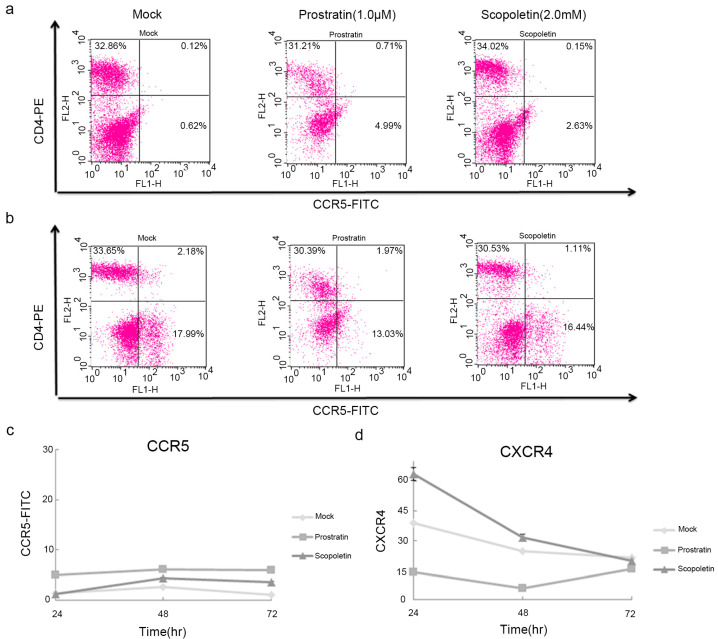
Effects of Scopoletin on the expression of CCR5 and CXCR4. (**a**) Human PBMCs were treated with Scopoletin (2 mM) or Prostratin (1 µM) for 72 h. (**b**) The expression of CCR5 and CXCR4 was detected by flow cytometry using antibodies against CCR5 and CXCR4, respectively. Fold change in the expression of (**c**) CCR5 and (**d**) CXCR4, as compared to mock treatment for different timepoints. The results are representative of three independent experiments.

**Figure 6 ijms-24-12649-f006:**
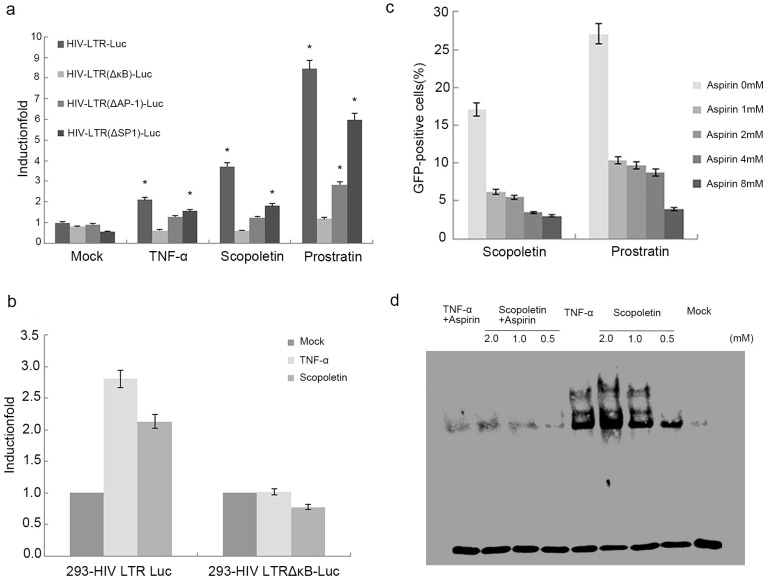
Scopoletin activates HIV-1 LTR through the induction of NF-κB. (**a**) HEK 293 cells were transfected with HIV1-LTR luc, HIV1-LTR (∆κB) luc, HIV1-LTR (∆AP-1) luc, and HIV1-LTR (∆SP1) luc. At 24 h post-transfection, the cells received mock treatment or treatment with Scopoletin (2 mM) TNF-α (10 ng/mL) and Prostratin(1 uM). Luciferase activity was measured after 24 h of treatment. The error bars indicate standard deviation. (**b**) The wild HIV-1 LTR-luc-neo and mutant HIV-1 LTR ∆κB-luc-neo cells were treated with Scopoletin (2.0 mM) and TNF-α (10 ng/mL). Luciferase activity was measured after 24 h of treatment. (**c**) C11 cells were pretreated with various concentrations of Aspirin for 3 h and subsequently treated with Scopoletin (2 mM) or Prostratin (10 ng/mL) for 24 h. The percentage of GFP+ cells was measured by flow cytometry. Data are presented as the mean ± standard deviation of three independent experiments. (**d**) Scopoletin stimulates nuclear NF-κB DNA binding. C11 cells were pretreated with or without Aspirin (10 mM) for 1 h and subsequently treated with Scopoletin at the indicated concentrations or TNF-α (10 ng/mL) for 3 h. Nuclear extracts were isolated and subjected to an electrophoretic mobility shift assay with biotin-labeled NF-κB enhancer DNA probes. * *p* < 0.05.

## Data Availability

All the data generated during the current study are included in the manuscript.

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
