# Peer review of "Scopoletin Reactivates Latent HIV-1 by Inducing NF-κB Expression without Global T Cell Activation"

_ijms, 2023, doi:10.3390/ijms241612649_

Round 1

Reviewer 1 Report

The study presented by the authors illustrates the prospective therapeutic effect of Scopoletin on HIV-1 through the reactivation of HIV-1 latency. Here are my suggestions to robust this study:

1.      The abstract should provide clear details about the study's methodology and its associated results. The current abstract appears to resemble a discussion section rather than a summary.

2.      Consistency is necessary in figure labeling; drug names should be represented uniformly, either in abbreviated form or in full.

3.      Consider adding methods such as the p24 antigen ELISA or qRT-PCR for measuring HIV-1 reactivation.

4.      It is necessary to clearly label the names of the cell lines in each individual graph of Figure 2.

5.      A justification is required for using different concentrations of Scopoletin in C11 and A10.6, especially when the difference is a mere 0.4mM. Generally, testing the EC50 and CC50 of Scopoletin on various T cells would aid in demonstrating its efficacy and cytotoxicity.

6.      Statistical significance needs to be clearly indicated in Figure 2.

7.      Figure 4 should also include measurements of additional T cell activation markers such as CD38 and HLA-DR, not just CD25 and CD69.

8.      Figure 6. D should explicitly indicate the dosage used.

9.      The effectiveness and cellular toxicity of combining Scopoletin with antiretroviral inhibitors should be evaluated in an HIV latency model.

nil

Author Response

The abstract should provide clear details about the study's methodology and its associated results.

A:Thank you for your comments on our manuscript. Those comments are very valuable and helpful for revising and improving our manuscript. We have gone through your comments carefully and made revisions accordingly which we hope will satisfy you. Note that changes made in the text are highlighted in red. Please see our point-by-point reply to comments made by the reviewers below.

  1. The current abstract appears to resemble a discussion section rather than a summary.

A:We thank the reviewer for this professional suggestion. We have improved our abstract in the revised manuscript.(Line 54-64)

  1. Consistency is necessary in figure labeling; drug names should be represented uniformly, either in abbreviated form or in full.

A:We are very sorry for our negligence. Now we have modified the figure labeling.

  1. Consider adding methods such as the p24 antigen ELISA or qRT-PCR for measuring HIV-1 reactivation.

A:Thank you very much for your suggestion. According to your suggestion, we have added the P24 antigen ELISA on A10.6 cell lines in the revised manuscript.(Figure 2e, Line 165-169,180-183,493-505)

  1. It is necessary to clearly label the names of the cell lines in each individual graph of Figure 2.

A:Thanks for your suggestion. We have labeled the names of the cell lines in Figure 2.

  1. A justification is required for using different concentrations of Scopoletin in C11 and A10.6, especially when the difference is a mere 0.4mM. Generally, testing the EC50 and CC50 of Scopoletin on various T cells would aid in demonstrating its efficacy and cytotoxicity.

A:Thank you for your suggestion. Setting the concentration to only double seems inappropriate. The 50% cytotoxic concentration (CC50) of Scopoletin was calculated in human primary PBMCs, Jurkat T and HEK 293 cells to further evaluate cytotoxicity. Scopoletin did not exhibit cytotoxicity in human primary PBMCs at any of the different concentrations. The CC50 in the HEK 293 and Jurkat T cells for Scopoletin was 15.18 and 17.366 mM. We should test CC50 of Scopoletin on various T cells.

  1. Statistical significance needs to be clearly indicated in Figure 2.

A:We have labeled the statistical significance in Figure 2.

  1. Figure 4 should also include measurements of additional T cell activation markers such as CD38 and HLA-DR, not just CD25 and CD69.

A:Thank you for your suggestion. We will study this content in future experiments.

  1. Figure 6. D should explicitly indicate the dosage used.

A:We have modified the Figure 6d.

  1. The effectiveness and cellular toxicity of combining Scopoletin with antiretroviral inhibitors should be evaluated in an HIV latency model.

A:Thank you for your suggestion. The experiments you mentioned need to be conducted in the P security laboratory, and these require appointment and application. We hope that the existing data can be shared with readers.

Reviewer 2 Report

The manuscript by Zhu et al. describes an investigation of the phytocompound Scopoletin as an activator of latent HIV. Their experiments employ 2 Jurkat-derived cell lines carrying latent HIV genomes and GFP to detect LTR promoter activity, HEK293 cells transfected with LTR-reporter plasmids, and an unspecified cell line stably expressing LTR reporters. They show that Scopoletin activates the HIV LTR and attribute this effect to activation of NF-kB.

This study is interesting but requires additional experiments and modifications to the text.

It would be informative to further evaluate the effects of Scopoletin on HIV reactivation by measuring the expression of the 3 classes of HIV transcripts.

Does Scopoletin affect the viability of the HIV-infected cell lines?

Experiments with LTR-reporters (Fig. 6a and b) should also be carried out using Prostratin to allow comparison of its effects with those of Scopoletin.

The involvement of the NF-kB pathway should be investigated using another inhibitor besides aspirin such as Bay-11-7082.

In the Discussion, the Authors should point out distinguishing properties of Scopoletin that may confer advantages over other described activating agents.

Additional comments

Title- it would read better in the present tense, i.e. replace reactivated with reactivates.

Line 47- replace has been with is (present tense).

Lines 83-85- and lines 286-289- cell lines A10.6 and C11 should be described in more detail with regard to the position of GFP in the viral genome and ability/inability to produce infectious virus. Does cell line A10.6 correspond to ARP-9849 in the AIDS Reagent Program? If so, this should be indicated in the Materials and Methods. The paper by Jordan et al. (57) should be cited on line 85.

The last part of the Introduction or the first paragraph of the results should provide a description of Prostratin. The paper by Kulkosky (ref. 50) as well as the studies of Prostratin by Korin WD et al. (2002) and Williams SA et al. (2004) should be cited in the description.

Lines 126-127- The text should also describe the results obtained with TNF-alfa alone.

The legend to Figures 3b-d indicate treatment with Prostratin as well as Scopoletin. This must be an error.

On Line 148, the treatment times for the 2 cell lines (48 or 72 hrs) could be omitted.

Line 149 should indicate the test used to measure cell viability i.e., an assay based on conversion of a tetrazolium salt to a formazan product (CCK-8 assay).

The chemical name for aspirin (acetyl salicylic acid) should be provided the first time the drug is mentioned in the text.

On line 197, do the Authors mean to say treatment instead of disposal?

On lines 200-203, what is the derivation of the LTR-Luc-neo cell lines? Jurkat? HEK293?

The Discussion would be improved by comparing the observations made for Scopoletin with a selection of the many other agents that have been tested as potential HIV reactivators.

Line 294 cites Bay-11-7082. The manuscript does not describe experiments with this drug (but should - see comment above).

Line 315- CellQuest is a BD product (not Macintosh).

Line 361- latent, not lantent.

In general the manuscript is well-written.

In addition to comments above, the following improvements would help.

Line 257- Global T cell activation can have deleterious effects.

Line 286- the A10.6 cell line ... was obtained from...

Lines 287-289- the C11 cell line should be more clear, avoiding mixing up 'it' with 'they'.

Author Response

The manuscript by Zhu et al. describes an investigation of the phytocompound Scopoletin as an activator of latent HIV. Their experiments employ 2 Jurkat-derived cell lines carrying latent HIV genomes and GFP to detect LTR promoter activity, HEK293 cells transfected with LTR-reporter plasmids, and an unspecified cell line stably expressing LTR reporters. They show that Scopoletin activates the HIV LTR and attribute this effect to activation of NF-kB.

This study is interesting but requires additional experiments and modifications to the text.

  1. It would be informative to further evaluate the effects of Scopoletin on HIV reactivation by measuring the expression of the 3 classes of HIV transcripts.

A:Thank you for your suggestion. We have performed the p24 antigen ELISA assay in the revised manuscript, which is is a gold standard approach. The ELISA assay is a very sensitive experimental method to illustrate the effects of Scopoletin on HIV reactivation. (Figure 2e, Line 165-169,180-183,493-505)

  1. Does Scopoletin affect the viability of the HIV-infected cell lines?

A:Thank you for your suggestion. In fact, in the course of our experiments, we found that latent HIV-1 in latently infected cells A10.6 were apoptotic when the Scopoletin concentration was 3.2mM. At the same time, the solubility of Scopoletin in water is less than 1mg/ml, and in the experiment, we dissolved it through DMSO, which will also produce cellular toxicity. In this study, we hope to mainly discuss the activation effect of Scopoletin on latent HIV and communicate the results with peers first. In the subsequent experiments, we will improve the dissolution method of Scopoletin and reduce the toxicity to cells.

  1. Experiments with LTR-reporters (Fig. 6a and b) should also be carried out using Prostratin to allow comparison of its effects with those of Scopoletin.

A:It’s a real meaningful suggestion to compare the effects of Scopoletin and Prostratin. We have compared the the effectivity between Scopoletin and Prostratin in the revise manuscript.(Line 298-302, 341,530)

  1. The involvement of the NF-kB pathway should be investigated using another inhibitor besides aspirin such as Bay-11-7082.

A:Thank you for your suggestion. BAY 11-7082 is an IκBα phosphorylation and NF-κB inhibitor,Due to the lack of stock, we will study this content in future experiments.

  1. In the Discussion, the Authors should point out distinguishing properties of Scopoletin that may confer advantages over other described activating agents.

A:Thank you for your suggestion. We have made the necessary modifications in the discussion.( Line 416-427)

Additional comments

Title- it would read better in the present tense, i.e. replace reactivated with reactivates.

(revised edition: Title)

Line 47- replace has been with is (present tense).

(revised edition: Line 72)

Lines 83-85- and lines 286-289- cell lines A10.6 and C11 should be described in more detail with regard to the position of GFP in the viral genome and ability/inability to produce infectious virus. Does cell line A10.6 correspond to ARP-9849 in the AIDS Reagent Program? If so, this should be indicated in the Materials and Methods.

(revised edition: Line 292, Line 294-297,Line 431-439)

The paper by Jordan et al. (57) should be cited on line 85.

(revised edition: Line 124)

The last part of the Introduction or the first paragraph of the results should provide a description of Prostratin. The paper by Kulkosky (ref. 50) as well as the studies of Prostratin by Korin WD et al. (2002) and Williams SA et al. (2004) should be cited in the description.

(revised edition: Line 110-114)

Lines 126-127- The text should also describe the results obtained with TNF-alfa alone.

(revised edition: Line 197-200 )

The legend to Figures 3b-d indicate treatment with Prostratin as well as Scopoletin. This must be an error.

(revised edition: Line 216-220)

On Line 148, the treatment times for the 2 cell lines (48 or 72 hrs) could be omitted.

(revised edition: Line 228)

Line 149 should indicate the test used to measure cell viability i.e., an assay based on conversion of a tetrazolium salt to a formazan product (CCK-8 assay).

(revised edition: Line 226-229)

The chemical name for aspirin (acetyl salicylic acid) should be provided the first time the drug is mentioned in the text.

(revised edition: Line 315)

On line 197, do the Authors mean to say treatment instead of disposal?

(revised edition: Line 297)

On lines 200-203, what is the derivation of the LTR-Luc-neo cell lines? Jurkat? HEK293?

A: The LTR-Luc-neo cell lines are based on the HEK293.

The Discussion would be improved by comparing the observations made for Scopoletin with a selection of the many other agents that have been tested as potential HIV reactivators.

(revised edition: Line 416-427)

Line 294 cites Bay-11-7082. The manuscript does not describe experiments with this drug (but should - see comment above).

(revised edition: Line309,447)

Line 315- CellQuest is a BD product (not Macintosh).

(revised edition: Line 474-475)

Line 361- latent, not lantent.

---Comments on the Quality of English Language:

(revised edition: Line 553)

In general the manuscript is well-written.

In addition to comments above, the following improvements would help.

Line 257- Global T cell activation can have deleterious effects.

(revised edition: Line 377-378)

Line 286- the A10.6 cell line ... was obtained from...

(revised edition: Line 433)

Lines 287-289- the C11 cell line should be more clear, avoiding mixing up 'it' with 'they'.

(revised edition: Line 436)

Reviewer 3 Report

In this study, Zhu et al. investigated the potential activity as a latency-reversing agent (LRA) of Scopoletin, a glycoside of a plant-derived hydroxycoumarin derivative. The results showed that the glycoside had the LRA effect in latently HIV-infected cell lines without causing global activation in PBMCs, making it a promising LRA.

However, I find it rather difficult to draw this conclusion from the experimental results presented here. The main points raised are listed below.

1. Scopoletin has already been reported to have anti-inflammatory activity in previous reports (for reference, see #1 below).

It also exhibits activity on the order of µM. Furthermore, its activity (at least partly) is reported to be due to NFkB inhibition (refs #2, 3). In this point of view, the conclusion that the LRA effect was due to NFkB activation by Scopoletin, as shown in this paper, is inconsistent with previous findings. If one assumes that the LRA effect occurs by the activation through NFkB pathway, the absence of CD69 and CD25 activation of PBMCs is quite unnatural. The dose of Scopoletin used in this experiment was of mM order, and the possibility of NFkB activation due to a large excess dose cannot be ruled out.

2. Even if the LRA effect by Scopoletin was caused by the NFkB activation at the dose of Scopoletin used in this study, it would lack practicality and clinical application would be difficult, considering a variety of other effects in humans.

3. several reports have already shown that the combination of two LRAs can suppress T-cell activation with high LRA activity, so the value of this substance is not high. Of a total of 63 cited references, only one report (reference 16) is from 2021 or later, and it is not appropriate that references that should be cited to reasonably evaluate this study were not cited.

#1: Antika, Lucia Dwi, Tasfiyati, Aprilia Nur, Hikmat, Hikmat and Septama, Abdi Wira. "Scopoletin: a review of its source, biosynthesis, methods of extraction, and pharmacological activities" Zeitschrift für Naturforschung C, vol. 77, no. 7-8, 2022, pp. 303-316. https://doi.org/10.1515/znc-2021-0193

#2: Pereira Dos Santos Nascimento MV, Arruda-Silva F, Gobbo Luz AB, Baratto B, Venzke D, Mendes BG, Fröde TS, Geraldo Pizzolatti M, Dalmarco EM. Inhibition of the NF-kB and p38 MAPK pathways by scopoletin reduce the inflammation caused by carrageenan in the mouse model of pleurisy. Immunopharmacol Immunotoxicol. 2016 Oct;38(5):344-52. doi: 10.1080/08923973.2016.1203929.

#3: Fan Z, Tang D, Wu Q, Huang Q, Song J, Long Q. Scopoletin inhibits PDGF-BB-induced proliferation and migration of airway smooth muscle cells by regulating NF-kB signaling pathway. Allergol Immunopathol (Madr). 2022 Jan 1;50(1):92-98. doi: 10.15586/aei.v50i1.517. PMID: 34965643.

Author Response

In this study, Zhu et al. investigated the potential activity as a latency-reversing agent (LRA) of Scopoletin, a glycoside of a plant-derived hydroxycoumarin derivative. The results showed that the glycoside had the LRA effect in latently HIV-infected cell lines without causing global activation in PBMCs, making it a promising LRA.

However, I find it rather difficult to draw this conclusion from the experimental results presented here. The main points raised are listed below.

  1. Scopoletin has already been reported to have anti-inflammatory activity in previous reports (for reference, see #1 below).

 It also exhibits activity on the order of µM. Furthermore, its activity (at least partly) is reported to be due to NFkB inhibition (refs #2, 3). In this point of view, the conclusion that the LRA effect was due to NFkB activation by Scopoletin, as shown in this paper, is inconsistent with previous findings. If one assumes that the LRA effect occurs by the activation through NFkB pathway, the absence of CD69 and CD25 activation of PBMCs is quite unnatural. The dose of Scopoletin used in this experiment was of mM order, and the possibility of NFkB activation due to a large excess dose cannot be ruled out.

A:  Thank your comments. We found that Scopoletin reactivated latent HIV-1 by inducing NF-κB expression, the conclusion that the LRA effect was due to NFkB activation by Scopoletin, is inconsistent with some previous findings, but there are also reports that this compound can activate the NF-κB pathway (     â‘ Kim EK, Kwon KB, Shin BC, Seo EA, Lee YR, Kim JS, Park JW, Park BH, Ryu DG. Scopoletin induces apoptosis in human promyeloleukemic cells, accompanied by activations of nuclear factor kappaB and caspase-3. Life Sci. 2005 Jul 1;77(7):824-36.    â‘¡Seo EJ, Saeed M, Law BY, Wu AG, Kadioglu O, Greten HJ, Efferth T. Pharmacogenomics of Scopoletin in Tumor Cells. Molecules. 2016 Apr 15;21(4):496.). The conflicting results may be related to different cell lines and drug doses.

  1. Even if the LRA effect by Scopoletin was caused by the NFkB activation at the dose of Scopoletin used in this study, it would lack practicality and clinical application would be difficult, considering a variety of other effects in humans.

A: As you mentioned that It exhibits activity on the order of µM, Indeed, it would be difficult to further clinical application .

  1. several reports have already shown that the combination of two LRAs can suppress T-cell activation with high LRA activity, so the value of this substance is not high. Of a total of 63 cited references, only one report (reference 16) is from 2021 or later, and it is not appropriate that references that should be cited to reasonably evaluate this study were not cited.

A: We are glad to receive your comments and thank you for your suggestion. About two LRAs suppress T-cell activation with high LRA activity, we have made the necessary modifications in the discussion. We have added some recent references. (Line 416-427)

Round 2

Reviewer 1 Report

The authors have addressed the concerns and suggestions and I agree to recommend the publication of their work.

Author Response

Thank you very much for your comments and professional advice. We really appreciate your efforts in reviewing our manuscript.

Reviewer 2 Report

Zhu et al. have addressed most of the comments made by this reviewer and have substantially improved the manuscript.

One minor issue regards the blunt manner in which the manuscript ends,   with a description of reactivators described by Gramatica et al. (lines 418-429). The ending would be improved by adding a sentence stating that it would be of interest to directly compare the effects of Scopoletin and the GSK-3 inhibitors in the Jurkat-derived cell lines and in patients’ samples.

Line 4 should be deleted.

Line 131- substitute ‘measure the expression of latent HIV-1’ with ‘detect reactivation of latent HIV-1’

The sentence on lines 202-204 is redundant with the previous sentence and could be deleted.

Lines 305 and 309- replace ‘wild’ with ‘wildtype’

Line 421- delete the initial ‘A’

Line 424- delete ‘were tested’

Line 437- ‘the GFP gene in place of the Nef gene’

Lines 438-440- The text would read better as follows:  cell line C11, also derived from Jurkat cells, expresses GFP as a marker of Tat-driven HIV LTR activity and carries mutations in the Env and Vpr genes.

Lines 499 and 500- replace ‘by with ‘with’. The remainder of the paragraph (lines 502-507) is written in the style of a protocol. It should read as a description.

Author Response

One minor issue regards the blunt manner in which the manuscript ends,   with a description of reactivators described by Gramatica et al. (lines 418-429). The ending would be improved by adding a sentence stating that it would be of interest to directly compare the effects of Scopoletin and the GSK-3 inhibitors in the Jurkat-derived cell lines and in patients’ samples.

A: Thank you for your precious comments and advice. Those comments are all valuable and very helpful for revising and improving our paper. We have modified the discussion in lines 418-439.

Line 4 should be deleted.

(revised editon: line 4)

Line 131- substitute ‘measure the expression of latent HIV-1’ with ‘detect reactivation of latent HIV-1’

(revised editon: line 131)

The sentence on lines 202-204 is redundant with the previous sentence and could be deleted.

(revised editon: line 201-202)

Lines 305 and 309- replace ‘wild’ with ‘wildtype’

(revised editon: line 305, 310)

Line 421- delete the initial ‘A’

(revised editon: line 421)

Line 424- delete ‘were tested’

(revised editon: line 430-432)

Line 437- ‘the GFP gene in place of the Nef gene’

(revised editon: line 448)

Lines 438-440- The text would read better as follows:  cell line C11, also derived from Jurkat cells, expresses GFP as a marker of Tat-driven HIV LTR activity and carries mutations in the Env and Vpr genes.

(revised editon: line 449-454)

Lines 499 and 500- replace ‘by with ‘with’.

(revised editon: line 510)

The remainder of the paragraph (lines 502-507) is written in the style of a protocol. It should read as a description.

(revised editon: line 508-521)

Reviewer 3 Report

The authors basically did not try to seriously address my concern, so I have no way to change my previous decision. 

OK

Author Response

Thanks for your valuable comments, we have added a discussion about the effect of Scopoletin on the NF-KB pathway in the Lines 417-423, and some recent researches in the References part marked in red. We hope that our modifications will be satisfactory to you.